

# Resilient dataset of rain clusters with life cycle evolution based on observations from the GPM DPR and Himawari-8 AHI

Aoqi Zhang[1], Chen Chen[2], Yilun Chen[1*], Weibiao Li[1], Shumin Chen[1], Yunfei Fu[3]

[1]School of Atmospheric Sciences, Sun Yat-sen University, and Southern Marine Science and Engineering Guangdong
Laboratory (Zhuhai), Zhuhai, 519082, China
[2]School of Applied Economics, Renmin University of China, Beijing, 100872, China
[3]School of Earth and Space Sciences, University of Science and Technology of China, Hefei, 230026, China

*Correspondence to*: Yilun Chen (chenylun3@mail.sysu.edu.cn)

**Abstract.** Our knowledge of the properties of precipitation and clouds over their life cycles has progressed with the rapid
development of satellite observations. However, previous studies have focused on the life cycle evolution of the macroscale
features of precipitation and clouds, whereas the evolution of the microphysical properties of precipitation and clouds over
their life cycles is yet to be determined. One of the reasons for this lack of knowledge is the fact that there is no single dataset
providing both the three-dimensional structure of precipitation and the relevant life cycle properties. We identified initial rain
clusters (RCs) from the Global Precipitation Measurement 2ADPR dataset and mesoscale convective systems (MCSs) from
the Himawari-8 Advanced Himawari Image gridded product. Based on the contours of the initial RCs and MCSs, we then
carried out a series of resilient processes, including filtration, segmentation, and consolidation, to obtain the final RCs. The
final RCs had a one-to-one correspondence with the relevant MCS. We extracted the RC area, central location, average radar
reflectivity profile, average droplet size distribution profile and other precipitation information from the final RCs and retrieved
the life cycle evolution of the MCS area, location, and cloud-top brightness temperature from the corresponding MCSs and
tracking algorithms. We provide both three-dimensional precipitation information and life cycle information in our resilient
dataset. This dataset facilitates studies of the life cycle evolution of precipitation and provides a good foundation for convection
parameterizations in precipitation simulations. The dataset used in this paper is freely available at
https://doi.org/10.5281/zenodo.5598418 (Zhang et al., 2021).

## 1 Introduction

The life cycle of clouds has a vital role in the atmospheric water cycle. The water resources of the Earth are constantly
replenished and a dynamic water balance is achieved through the formation, movement, precipitation, and dissipation processes
of clouds (Oki and Kanae, 2006; Li et al., 2020). The scale and density of cloud particles are constantly changing during the
life cycle of clouds as a result of the influence of environmental factors, such as the atmospheric temperature, water vapor
content, aerosols, and atmospheric movements (Rosenfeld et al., 2018; Chen et al., 2020b). If the scale and density of cloud
particles increase to a certain level, then the cloud particles are likely to collide frequently with each other in random motion,



forming larger precipitation droplets (Freud and Rosenfeld, 2012; Houze, 2014). Precipitation droplets fall to the ground through complex microphysical processes such as deposition growth, riming growth, rime splintering, aggregation growth, melting, and evaporation (Morrison and Milbrandt, 2015; Aggarwal et al., 2016). The accompanying release or absorption of latent heat are important for regulating regional and global energy budgets (Jung et al., 2011; Min et al., 2013; Nelson and

L'Ecuyer, 2018). Parameterization schemes for precipitation and its life cycle evolution are still difficult questions in cloud models (Chawla et al., 2018; Freitas et al., 2018; Gentine et al., 2018). Revealing the relationships between three-dimensional precipitation microphysics and cloud life cycle evolution will improve our physical cognition of clouds and precipitation and provide a factual basis for the parameterization of precipitation.

The rapid development of satellite remote sensing in the 21st century has brought new opportunities for studies of the life cycle

evolution of clouds and precipitation. Space-borne active radar systems, such as the Tropical Rainfall Measuring Mission (TRMM) Precipitation Radar (PR) and the Global Precipitation Measurement (GPM) Dual-frequency Precipitation Radar (DPR) provide stereo observations of precipitation structure and microphysics (Miura et al., 2012; Iguchi et al., 2012). The high spatiotemporal resolution of visible and infrared observations from the new generation of geostationary satellites, such as the Himawari-8 and FY-4A satellites, provides a robust guarantee for identifying and tracking mesoscale convective systems

(MCSs) or clouds (Vant-Hull et al., 2016; Wall et al., 2018; Chen et al., 2020b; Zhuge and Zou, 2018; Zhang et al., 2021). The coordinated application of observations from geostationary satellites and space-borne active radar systems provides the conditions necessary to reveal the life cycle evolution of clouds and precipitation.

Based on coordinated applications, scientists have carried out many studies on the life cycle evolution of precipitating clouds. Filleau and Roca (2013b) combined TRMM PR and geostationary satellite data and showed that the precipitation intensity was

strongest at about one-third of the cloud life cycle, whereas the proportion of convective pixels gradually decreased with the development of the clouds. Based on geostationary satellite observations and precipitation data from the Climate Prediction Center morphing technique (CMORPH), Ai et al. (2016) found that the lowest cloud-top temperature does not correspond to the heaviest precipitation during the life cycle of an MCS. The MCS tends to produce the heaviest precipitation first and then the minimum brightness temperature. By combining geostationary satellite observations and ground-based radar data, Chen et

al. (2020a) showed that the development and dissipation stage of MCSs is significantly longer than the mature stage and that the proportion of anvils in the cloud gradually increases during the life cycle, whereas the proportion of convex cores gradually decreases. Using Integrated Multi-satellitE Retrievals for GPM (IMERG) and geostationary satellite observations, Li et al. (2021) revealed the semi-diurnal cycle of deep convective systems over Eastern China.

These studies have improved our understanding of the life cycle evolution of macroscale features of precipitating clouds.

However, there is still a lack of statistical studies of the life cycle evolution of precipitation microphysics processes. Our previous study combined the droplet size distribution (DSD) from the GPM DPR and from the Himawari-8 geostationary satellite dataset for April–August 2016 and constructed a microphysical conceptual model of precipitating clouds at different life stages over eastern China (Zhang and Fu, 2018). Kumar et al. (2020) showed the vertical structure of the precipitating



system at different cloud life stages in the mountains of the Andes. Research is limited by the large amount of satellite data
and complex processing algorithms.

To facilitate precipitation studies, scientists have developed multiple datasets based on the orbital precipitation product TRMM
PR 2A25 and the subsequent GPM 2ADPR. Liu et al. (2008) constructed an event-based precipitating system dataset by
grouping the contiguous precipitating area detected by the TRMM PR; the dataset was later extended to the GPM 2ADPR (Liu
and Zipser, 2015). This dataset has received widespread attention (Houze et al., 2015; Aggarwal et al., 2016; Nishant et al.,
2019; Schumacher and Rasmussen, 2020) because it can effectively reduce the complexity of event-based precipitation
research. More efforts have been made to merge datasets from different instruments onboard the TRMM, including PR 2A25,
the Visible Infrared Radiometer Scanner (VIRS) 1B01, the TRMM Microwave Image (TMI) 1B11 and environmental
information (Wilheit et al., 2009; Fu et al., 2013; Chen and Fu, 2017; Sun and Fu, 2021).

However, there is still no single space-borne dataset that can provide both event-based precipitation information and its
corresponding life cycle evolution. The initial rain clusters (RCs) identified from precipitation radar and the MCSs from
geostationary observations do not have a one-to-one correspondence, so we need to carry out a series of resilient reprocessing
algorithms to remove redundant information.

The paper consists of four parts. Section 2 describes the basic information in the dataset. Section 3 demonstrates the processing
algorithms, including the resilient reprocessing algorithms (Sections 3.1 and 3.2) and the MCS tracking algorithm (Section
3.3). Section 4 summarizes our dataset.

## 2 Data and methods

### 2.1 GPM 2ADPR dataset

The GPM core observatory carrying the DPR and the GPM Microwave Imager was launched on February 28, 2014. The GPM
DPR is the first space-borne dual-frequency precipitation radar and covers the globe from 65° S to 65° N. The GPM DPR
consists of a Ku-band (13.6 GHz) and a Ka-band (35.5 GHz) precipitation radar and operates on three different scan modes,
including the Ku-band Normal Scan (NS), the Ka-band Matched Scan (MS) and the Ka-band High-sensitivity Scan (HS). The
HS mode (24 beams) was changed on May 21, 2018 to match the outer swarth of the NS mode, whereas the MS mode (25
beams) is matched with the inner swath of the NS mode (Iguchi et al., 2018). The difference between the matched beams is
now about 30 m at nadir, whereas it was 300 m before May 21, 2018. The relevant minimum detectable reflectivity values for
the NS, MS and HS modes are 14.5, 16.7 and 10.2 dBZ, respectively (Hamada and Takayabu, 2016). Based on the three scan
modes, the official GPM orbital dataset provides three single-frequency products and a dual-frequency product (2ADPR). We
used the 2ADPR product, which provides the rain type, storm-top height, corrected reflectivity profile, DSD profile, rain rate
profile and other information at a horizontal resolution of 5 km and a vertical interval of 125 m.



## 2.2 Himawari-8 gridded product

The Himawari-8 satellite, equipped with the Advanced Himawari Image (AHI), was launched in 2016. The AHI operates at 16 visible and infrared wave bands from 0.46 to 13.3 μm. The spatial resolution of the Himawari-8 full-disk data varies with the wave band and includes 0.5, 1 and 2 km (Bessho et al., 2016). By performing a preliminary investigation of the infrared channel measurements from the AHI, Da (2015) showed that the sums of the observational and model error variance are about 1.5 K for the 6.2–7.3 μm channels and about 1 K for other infrared channels. We used the Himawari-8 full-disk product on

(0.05° × 0.05° grids) (ftp.ptree.jaxa.jp) for consistency with the data resolution of the GPM 2ADPR. We used the 10.4 μm brightness temperature at a temporal interval of 1 h.

## 2.3 Basic information for the resilient dataset

The derived resilient dataset of RCs with life cycle evolution covers eastern Asia from 2016 to the present day (Fig. 1). The center of the initial RC derived from the 2ADPR product is restricted to the spatial range (90–150° E, 10–50° N). We tracked

the corresponding MCS over a wider spatial range (80–170° E, 0–60° N) because the central area is in the region of subpolar westerlies with strong high-level westerly winds.

Figure 2 shows the processing flowchart for the resilient dataset. Following the RC identification method in Fu et al. (2020), we identified continuous precipitation pixels (>0 mm h$^{-1}$) in the GPM 2ADPR orbital data as initial RC. All the initial RCs were temporarily retained, regardless of the area of the RC and whether the RC was affected by truncation of the DPR swath.

This RC identification method has been widely used in event-based precipitation and cloud research (Feral et al., 2000; Nesbitt et al., 2006; Chen et al., 2017; Zhang et al., 2018). Using a similar method and a brightness temperature threshold of <235 K (following Mapes and Houze, 1993), we also identified MCSs from the Himawari-8 AHI 10.4 μm brightness temperature data. The contours of the initial RCs were very different from those of the MCSs as a result of the randomness of precipitation, the mismatch between cold cloud-top and near-surface precipitation, the temporal difference between the two datasets, the

truncation of the DPR swath and many other factors. We therefore applied a series of resilient reprocessing algorithms to give a better correspondence between the RCs and the MCSs.

We compared the contours of the initial RCs and the MCSs to determine the mapping relationships between them. Specifically, we gridded the DPR pixels to (0.05° × 0.05°) grids (consistent with the Himawari-8 data) and determined the overlapping grids between the initial RCs and the MCSs at the nearest time (±30 min). If there was no overlapping grid between one initial RC

and any MCS, then the initial RC was matched to the nearest MCS within 100 km. If there was no MCS within 100 km of the initial RC, then the initial RC was filtered out. These filtered RCs contain isolated warm rain over the ocean, with a low rain-top and usually weak near-surface precipitation (Lau and Wu, 2011; Chen and Fu, 2017).

After filtration, the remaining RC corresponded to at least one MCS. We carried out other resilient processes, including segmentation and consolidation, on individual clusters to derive the final RCs that corresponded to the relevant MCSs. The

specific processes of segmentation and consolidation are described in the following section. We tracked the corresponding



MCSs forward and backward to derive the life cycle evolution of the precipitating cloud. The tracking algorithm is described in the following section.

## 3 Applications

### 3.1 Resilient segmentation of RC

If one of the remaining RC corresponded to multiple MCSs, then it was resiliently segmented according to the contours of the MCSs. Figure 3 shows a precipitation event captured by the GPM DPR, which occurred at 10:08 UTC on June 1, 2016 over eastern China. The near-surface rain rate was mainly distributed in the interval 0.5–5 mm h$^{-1}$ (Fig. 3a). The region of low-brightness temperature was scattered over several small areas (Fig. 3b). The horizontal distribution of the initial RCs showed that the precipitation pixels within 32–35° N belonged to the same initial RC with an area of about 28000 km$^2$ (Fig. 3c). This

large initial RC had irregular boundaries and overlapped with seven different small MCSs with areas ranging from 75 to 1225 km$^2$ (Fig. 3c and 3d).

The segmentation algorithm consists of two main steps. We gridded the DPR pixels to (0.05° × 0.05°) grids to obtain the overlapping grids between the initial RCs and the MCSs. The first step was to restore the overlapping grids to the DPR pixels; the areas overlapping with different MCSs were marked as different RC cores. Figure 4a shows the RC cores after the first

step of the segmentation algorithm. Seven different RC cores are marked with different colors.

The second step was to use the image corrosion method (Gonzalez and Thomason, 1978) to gradually allocate the other DPR pixels within the initial RC to the RC cores. Specifically, we let the RC cores gain weight round by round until all the other DPR pixels were allocated. If, at a certain round of the corrosion process, one DPR precipitation pixel was allocated to multiple RCs cores, then we relocated this pixel following the principle of the smallest rain rate gradient. Figure 4b shows that the

initial RC was segmented into seven new RCs. The largest three new RCs had areas of 7325 km$^2$ (blue), 9850 km$^2$ (cyan) and 6225 km$^2$ (orange).

Figure 4c and 4d show the average DSD profiles of the three largest RCs. In general, all the average profiles of the droplet density ($dBN_w$) and the effective diameter ($D_m$) of the three RCs show a clear turning point at 5.5 km height (around the freezing level). This is because the lower layer of this precipitation event was very humid (not shown) and the precipitation

microphysics within this event was dominated by the "warm rain" process. However, the average near-surface $dBN_w$ of the southerly blue RC reached 37.4, which is significantly higher than the other two (36 and 35.7); the average near-surface $D_m$ of this southerly blue RC was about 1.1 mm, which was significantly smaller than the other two (about 1.15 mm). This suggests that there are significant differences in the precipitation microphysics inside these three RCs, so resilient segmentation of the RC is required.





## 3.2 Resilient consolidation of RCs

After the segmentation process, each RC had only one corresponding MCS. However, there may still be multiple RCs corresponding to one MCS. For consistency, we wanted to consolidate these multiple RCs into one single RC.

Figure 5 shows a precipitation event captured by the GPM DPR that occurred at 23:09 UTC on June 2, 2020 over the East China Sea during the Meiyu period. The horizontal distribution of the 10.4 μm brightness temperature shows that the Meiyu clouds extended northeasterly from Shanghai to South Korea (Fig. 5b). The corresponding MCS of this event consisted of two parts: a southerly near-circular MCS and a northly elongated MCS (Fig. 5d). The southerly near-circular MCS mainly corresponded to a large RC with smooth boundaries and intense precipitation (Fig. 5a and 5c). The central near-surface rain rate within the large RC exceeded 10 mm h$^{-1}$. The northerly elongated MCS corresponded to multiple small initial RCs with irregular boundaries and weak precipitation (Fig. 5a and 5c). The near-surface rain rate of the RCs was mostly weaker than 1.5 mm h$^{-1}$.

Following the resilient consolidation principle, the small segmented RCs corresponding to the northerly elongated MCS were consolidated into one new RC (Fig. 6a and 6b). The average DSD profiles of the main RCs before and after consolidation are presented in Fig. 6c and 6d. The precipitation microphysics were dominated by ice-phase processes. Above the frozen layer, the droplet size increased with decreasing height due to the deposition, riming and aggregation growth of droplets (Chen et al., 2020c). Below the frozen layer, the droplet size gradually decreased with decreasing height as a result of evaporation. The near-surface $dBN_{\mathrm{w}}$ was around 33 and the near-surface $D_{\mathrm{m}}$ was about 1.02 mm, indicating typical stratiform precipitation (Bringi et al., 2006; Wen et al., 2016). Specifically, the DSD profiles of the main RCs were fairly similar, proving that the consolidation process was reasonable.

By carrying out these resilient processes of filtration, segmentation, and consolidation on the initial RC, we obtained the final RC that had a one-to-one correspondence with the relevant MCS.

## 3.3 Life cycle evolution of the MCS

The life track of one final RC was derived from its corresponding (one-to-one) MCS. We took the corresponding MCS as the origin and tracked forward and backward from the MCS at temporal intervals of 1 h. The MCS tracking algorithm followed the widely used areal overlapping method with speed correction (Machado and Laurent, 2004; Filleau and Roca, 2013a; Ai et al., 2016; Chen et al., 2019; Wall et al., 2020). The threshold of the areal overlapping ratio was set to 50%—that is, the overlapping area of successive MCSs must be >50% of the area of the MCS at both later and earlier times. We first evaluated the moving speed of the MCS in the study area based on a preliminary tracking result without speed correction.

The movement of the MCS relied highly on the latitude and month as a result of the variance of the high-level wind field (Feng et al., 2021). Figure 7 shows the derived relationships between the moving speed of the MCS and latitude from April to June. In general, the zonal speed of the MCS was larger than the meridional speed, but their standard deviations were similar. The high-level wind field was weak in the region 0–10° N, which is affected by tropical depressions, so the average zonal and





meridional velocities of the MCS were close to 0 (Fig. 7a and 7b). To the north of 10° N, the average zonal velocity of the MCS was seen as an eastward movement affected by high-level westerly winds, whereas the average meridional velocity was seen as a northward movement (Fig. 7a and 7b). The average zonal speed of the MCS reached a peak at around 30° N, corresponding to the position of the western Pacific subtropical high (Zhang et al., 2020).

The average moving speed of MCSs were similar in different months from April to June (Fig. 7a and 7b), but their standard deviations showed clear differences (Fig. 7c and 7d). In the area at about 40° N, the standard deviations of the zonal and meridional speeds in April reached 1.4° E h⁻¹ and 1° N h⁻¹, respectively. By contrast, the standard deviations of the zonal and meridional speeds in June were about 0.5° E h⁻¹ and 0.4° N h⁻¹, respectively. The reason for the differences in these standard deviations requires further study, but may be related to the northward movement of the western Pacific subtropical high before the onset of the Meiyu season (Li et al., 2019). Besides, the movement of the MCS was fairly consistent during the Meiyu season, which usually occurs in June, as a result of the influence of a quasi-stationary front; the standard deviation of the moving speed of MCS would therefore be small.

We used the average velocity of the MCS ($\bar{r}$) as the initial velocity of the formal tracking algorithm. The standard deviation of the velocity of MCS was $\mu_r$. The forward tracking algorithm (similar for backward tracking) was as follows:

1) Assuming the MCS at time 0 to be set $\{A_0\}$, use the average velocity $\bar{r}$ to calculate the possible set $\{A'_t\}$ ($t = 1$) of the MCS at time 1.

2) Use $\{A'_t\}$ and the areal overlapping method to determine the actual MCS set $\{A_t\}$ at time $t$. If $\{A_t\}$ does not exist, stop tracking.

3) Calculate the instantaneous moving velocity of MCS ($r$) from $\{A_t\}$ and $\{A_{t-1}\}$. If $r$ exceeds the range of $[\bar{r} - \mu_r, \bar{r} + \mu_r]$, let $r = \bar{r}$.

4) Use the instantaneous velocity $r$ and $\{A_t\}$ to calculate the possible set $\{A'_{t+1}\}$ of the MCS at time $t+1$.

5) Let $t = t + 1$ and skip to step 2.

Figure 8 shows an example of the tracking algorithm. The precipitation event occurred at about 13:00 UTC on June 20, 2016 over Kyushu Island, Japan (Fig. 8d). The gray and light blue areas represent the estimated MCS $\{A_t\}$, whereas the deep blue and light blue areas represent the actual MCS $\{A'_t\}$. In this case, the estimated MCS showed good agreement with the actual MCS, with the areal overlapping ratio reaching 80% at most times (Fig. 8). During the eastward to northeastward movement of the MCS, the shape of the MCS gradually elongated along the direction of movement. Fig. 8a and 8b show that the area of the MCS changed dramatically when splitting. We therefore do not recommend the use of only the temporal variance of the MCS area to identify the life stage of MCSs.

Fig. 9a shows the entire moving track of this MCS. The MCS had a lifetime of 34 h with a moving track >2000 km. The MCS originated on the ocean about 50 km west of Kyushu Island, Japan and then moved eastward for about 10 h. After reaching Shikoku Island, the direction of movement turned northeastward and the MCS lasted for another 24 h until it dissipated on the ocean about 1000 km east of Honshu Island.





Fig. 9b and 9c show the temporal variations in several important parameters of the MCS. During the life cycle, the area of the MCS first increased and then decreased over time; the peak MCS area of 250,000 km$^2$ occurred at about one-third of the life cycle (Fig. 9b). The moving speed of the MCS ranged from 30 to 120 km h$^{-1}$, but was about 60 km h$^{-1}$ during most of the life cycle (Fig. 9b). The temporal variations in the average and minimum 10.4 μm brightness temperature showed first a sharp decrease and then a slow increase (Fig. 9c). At the origin time, the brightness temperature of the MCS was in the trough period

and the area was increasing, suggesting that this precipitation event was caused by a mature stage MCS.

**4 Data availability**

The resilient dataset of rain clusters with life cycle evolution is freely available at https://doi.org/10.5281/zenodo.5598418 (Zhang et al., 2021).

**5 Discussion and conclusions**

We constructed a resilient dataset of rain clusters with life cycle evolution based on observations from the GPM DPR and Himawari-8 AHI. The three-dimensional precipitation structure of the RC was provided by GPM DPR data and the relevant life cycle evolution of the MCS was obtained from the Himawari-8 AHI. The purpose of this dataset is to facilitate three-dimensional studies of the life cycle evolution of precipitation. In the process of constructing the dataset, we used a series of satellite data processing methods as summarized in the following.

First, using a connected component analysis method, we extracted initial RCs and MCSs from GPM 2ADPR orbital data with a horizontal resolution of 5 km and Himawari-8 AHI 10.4 μm channel hourly grid data on (0.05° × 0.05°) grids. The contours of the initial RCs and MCSs were different a result of factors including the randomness of precipitation, the mismatch between cold cloud top and near-surface precipitation, the time differences between the two datasets and the truncation effect of the DPR swath.

Second, we performed a series of resilient reprocessing steps to remove redundant contour information between the initial RCs and the MCSs, including filtration, segmentation, and consolidation on the initial RCs. We obtained the final RCs, which corresponding one-to-one to the relevant MCSs. To evaluate the effectiveness of these algorithms, we carried out two case studies on the segmentation and consolidation algorithms. The results showed that the internal areas within the final RCs obtained from the resilient reprocessing had relatively consistent DSD profiles, indicating that they had similar precipitation

microphysics and that our reprocessing algorithms were reasonable and necessary.

Third, we tracked the relevant MCS, corresponding one-to-one to the final RC, both forward and backward to obtain the life cycle evolution. The specific tracking algorithm is given in the main text and is based on the areal overlapping method with speed correction. The case study of the tracking algorithm showed that the temporal evolution of the cloud-top brightness





temperature was more suitable for identifying the cloud life stages than the temporal evolution of the area of the MCS. The

shape of the MCS gradually elongated along the direction of movement during the life cycle of the MCS.

We then calculated the area, location, mean corrected radar reflectivity, mean DSD profiles, mean storm-top height, mean near-surface rain rate and other precipitation information of the final RC and the temporal evolution of area, location, cloud-top brightness temperature and other information of the MCS from the tracking results. Both the RC information and the life cycle information of relevant MCS were stored in the resilient dataset.

This new dataset greatly reduces the size of the dataset from >200 GB per month to <10 MB per month and avoids complex data processing algorithms, which will facilitate studies of event-based precipitation and its life cycle evolution. The cloud parameters with vertical revolution retrieved from Himawari-8 AHI will be added to the dataset to investigate the relationships between cloud microphysics, precipitation microphysics and cloud life cycle further (Chen et al., 2020b). This work is now in progress, but is not reported here due to the limited length of the paper. We will also develop and optimize our algorithms to

further improve our dataset, such as using Himawari-8 data with a higher temporal resolution and longer time periods, and segmenting the MCS based on convective cores.

**Author contributions**

A.Z. and Y.C. conceived the idea and proposed this study. A.Z. prepared the data and drafted the manuscript. All the authors discussed the concepts and edited the manuscript.

**Competing interests**

The authors declare no competing interests.

**Acknowledgements**

We are greatly appreciative of the National Aeronautics and Space Administration (NASA) for providing GPM 2ADPR dataset and the Japan Meteorological Agency (JMA) for providing Himawari-8 data.

**Financial Support**

This work was supported by the National Natural Science Foundation of China (Grant Nos. 42005062, 42075004, and 42105068), the Guangdong Basic and Applied Basic Research Foundation (Grant No. 2021A1515011404), the Fundamental Research Funds for the Central Universities Sun Yat-sen University (Grant number 2021qntd29), the Innovation Group Project of Southern Marine Science and Engineering Guangdong Laboratory (Zhuhai) (Grant number 311021009), and the

Fundamental Research Funds for the Guangzhou Science and Technology Plan Project (Grant number 20193010036).





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

**Figure captions**

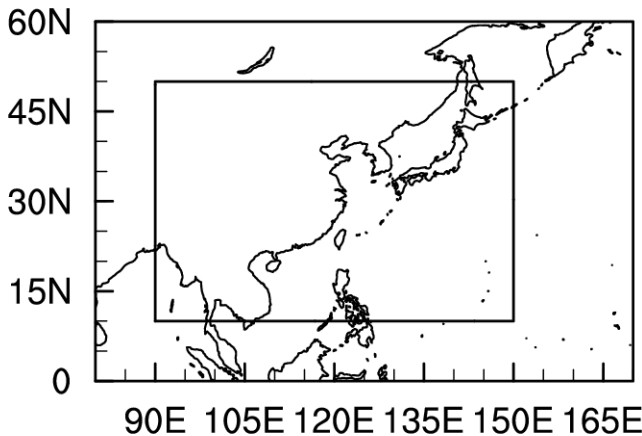

**Figure 1: Horizontal coverage of our dataset. The black rectangle represents the coverage of the RC centers.**








**Figure 2: Processing flowchart for our dataset.**

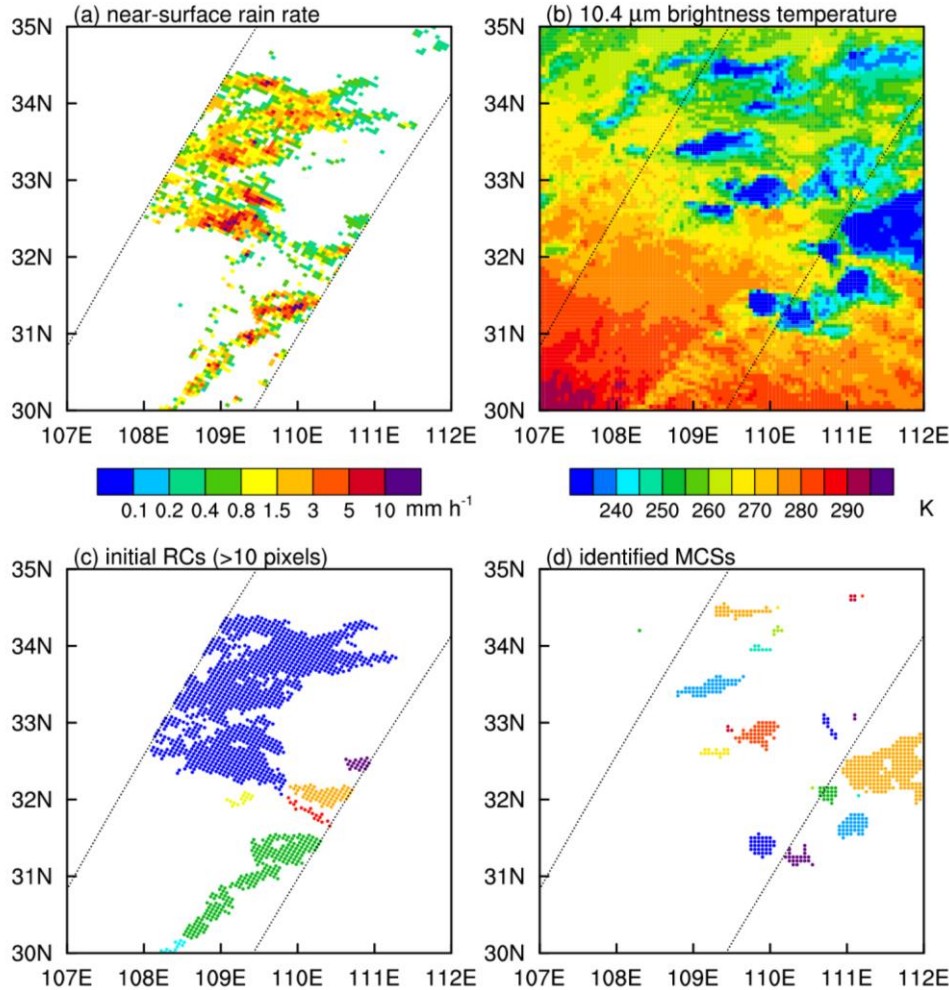

**Figure 3: Horizontal distributions of the (a) near-surface rain rate, (b) 10.4 µm brightness temperature, (c) initial RCs and (d) identified MCSs for the precipitation event occurring at 10:08 UTC on June 1, 2016. The adjacent dots of different colors in parts (c) and (d) indicate different RCs or MCSs.**




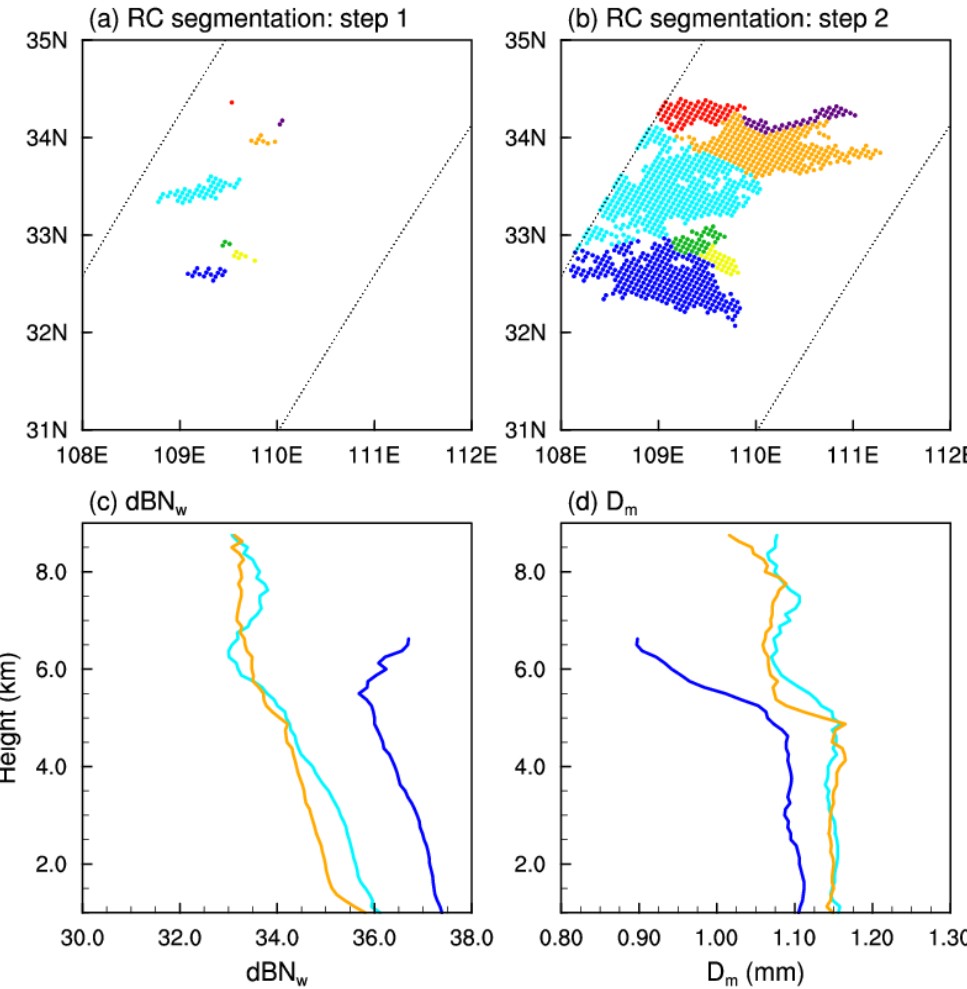

**Figure 4: (a, b) Demonstrations of the two steps in RC division for the RC occurring at 10:08 UTC on June 1, 2016 and (c, d) the average DSD profiles of the divided RCs. (c, d) The colors of the DSD profiles corresponding to the colors of the RCs shown in panel (b).**

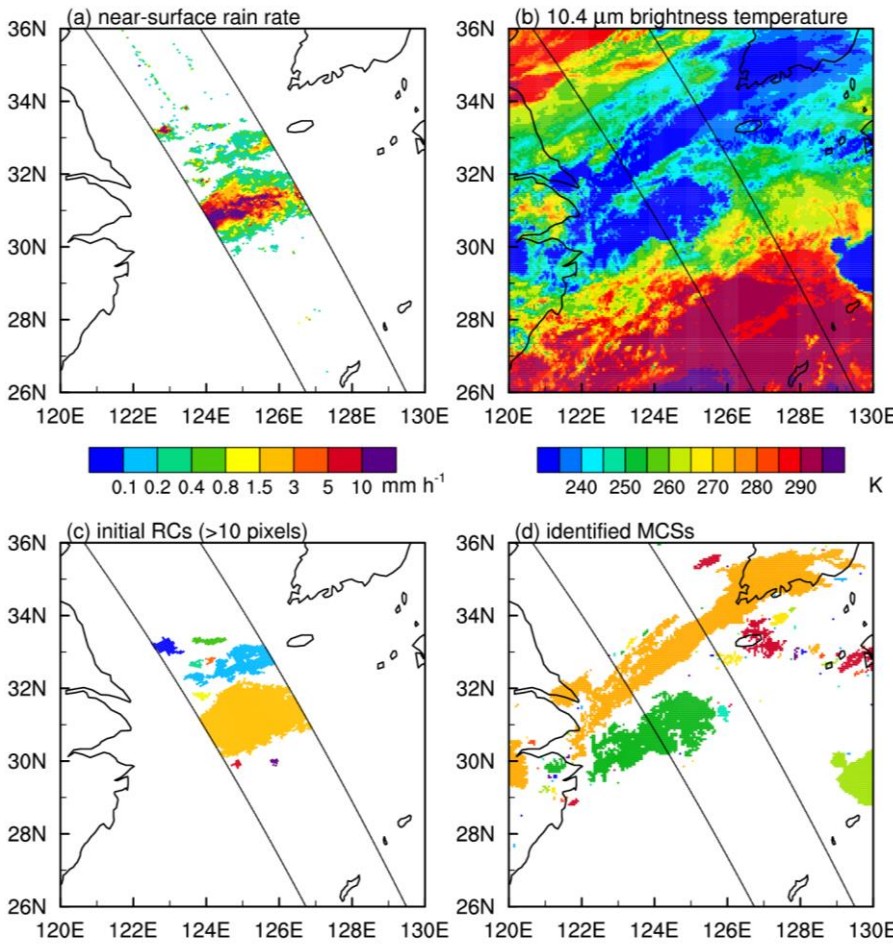


**Figure 5: Horizontal distributions of the (a) near-surface rain rate, (b) 10.4 μm brightness temperature, (c) initial RCs and (d) identified MCSs for the precipitation event occurring at 23:09 UTC on June 2, 2020. The adjacent dots of different colors in parts (c) and (d) indicate different RCs or MCSs.**

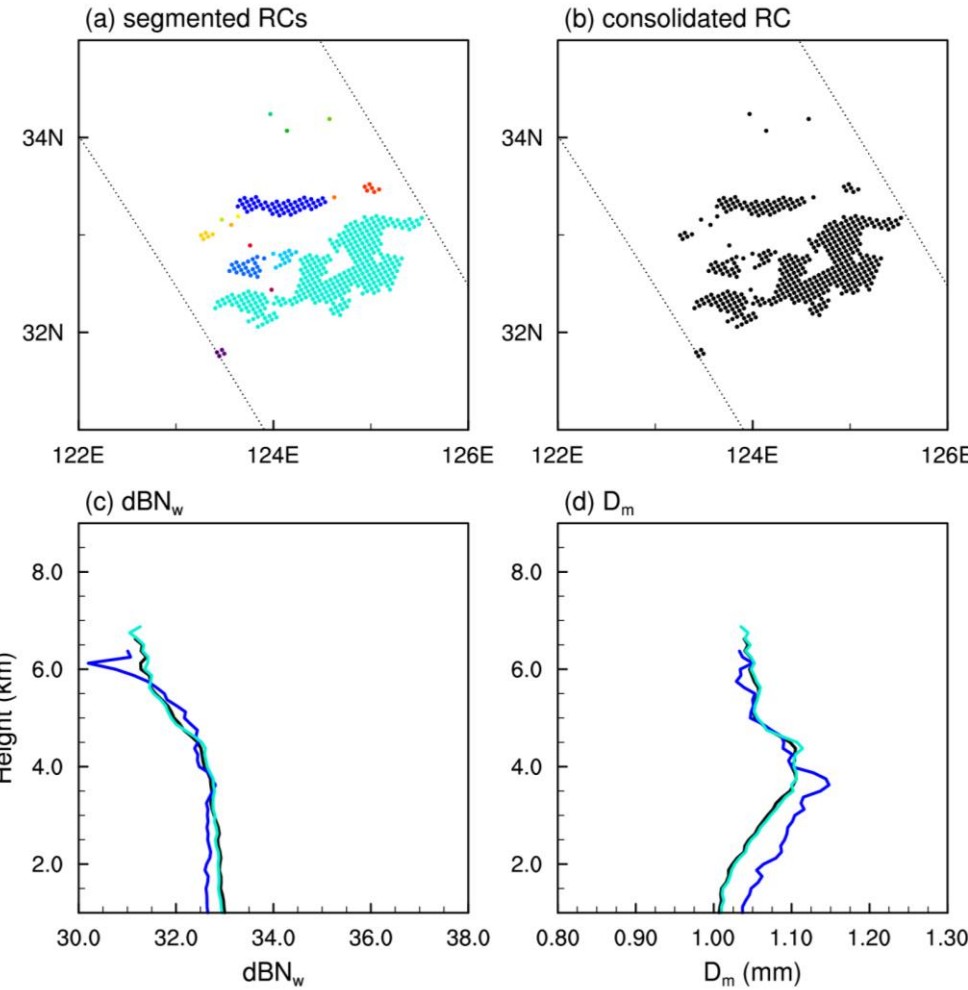

Figure 6: Horizontal distributions of (a) the segmented RCs and (b) the consolidated RC, and average DSD profiles (c-d) of them.



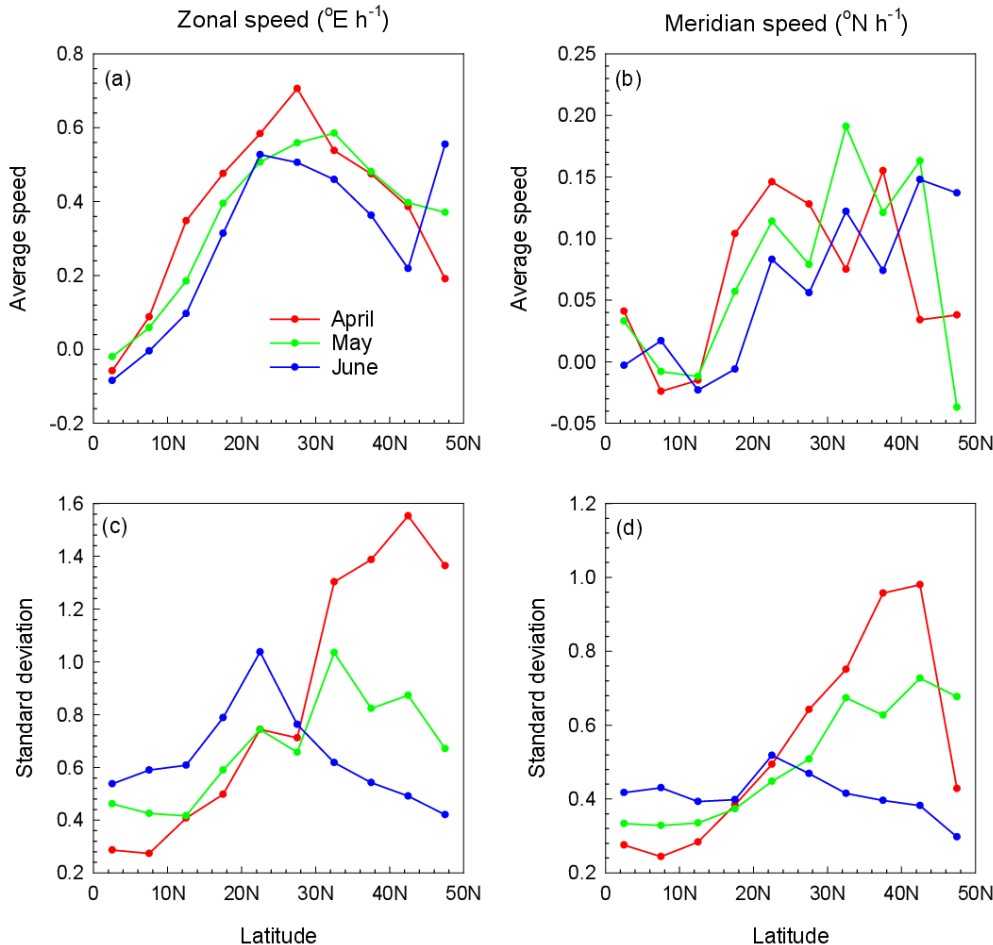

**Figure 7: (a, b) Average zonal and meridian moving speed of MCSs and (c, d) their standard deviations derived from the preliminary tracking result.**

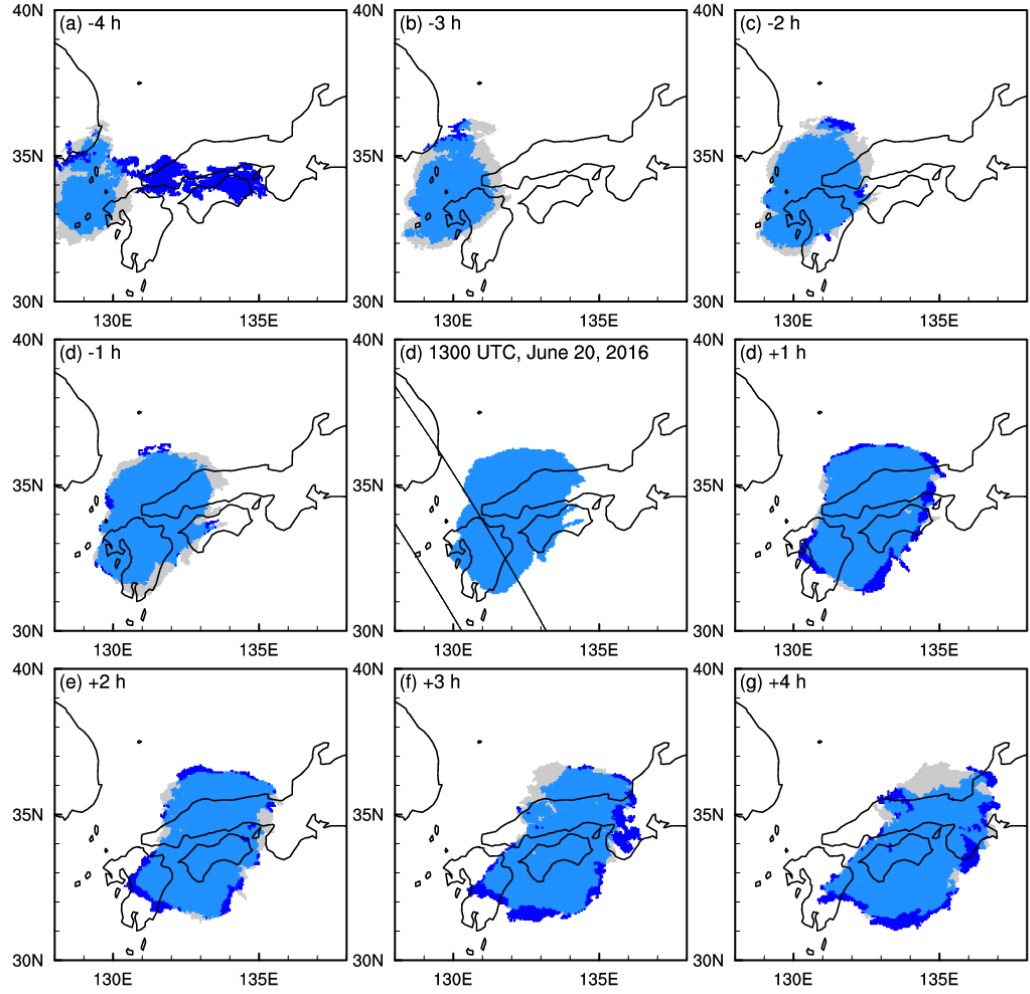


**Figure 8: Demonstration of the tracking process for the precipitation event occurring at about 13:00 UTC on June 20, 2016. The light blue and deep blue areas indicate the actual MCS. The gray and light blue areas indicate the calculated MCS from the last record.**



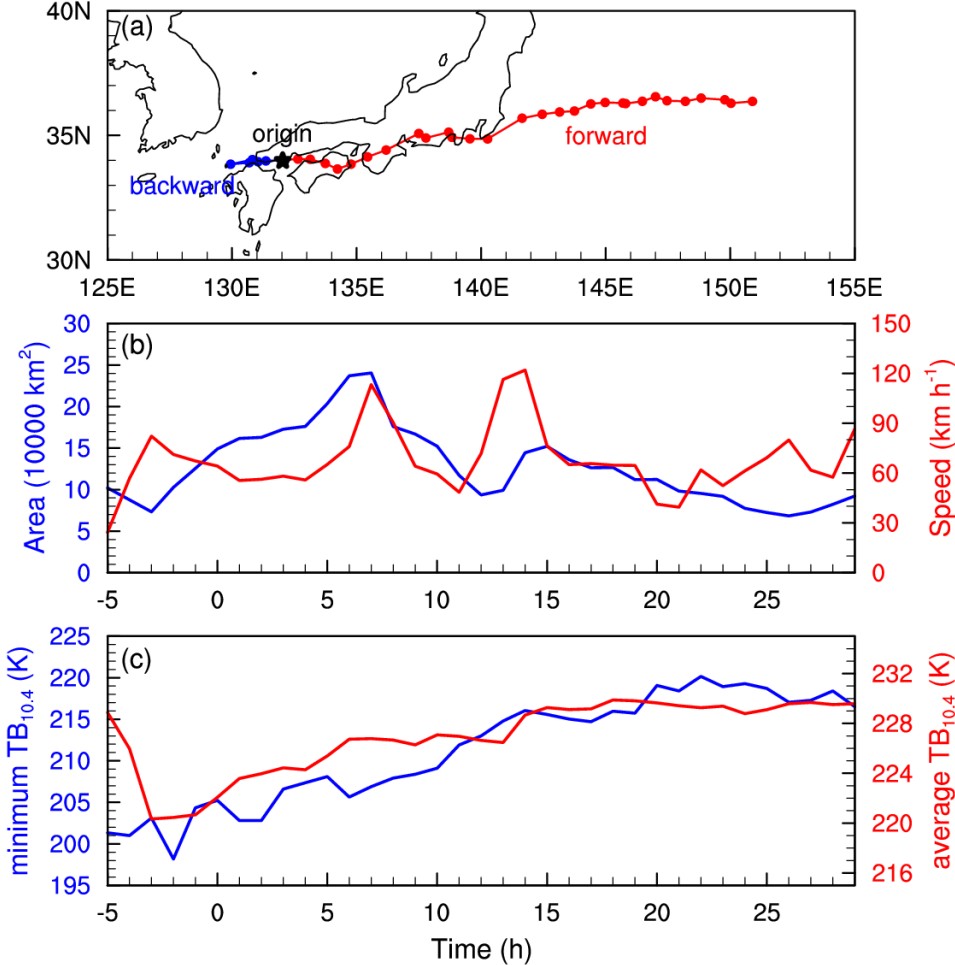

**Figure 9: (a) Moving track of the MCS occurring at 13:00 UTC on June 20, 2016 and (b) temporal variations in the area, speed, and (c) minimum and average 10.4 μm brightness temperature of the MCS.**