# Peer review of "Resilient dataset of rain clusters with life cycle evolution during April to June 2016–2020 over eastern Asia based on observations from the GPM DPR and Himawari-8 AHI"

_Earth System Science Data, 2021_

## Author Comment (AC1)

**Comment on essd-2021-298**

**Anonymous Referee #1**

Referee comment on "Resilient dataset of rain clusters with life cycle evolution based on observations from the GPM DPR and Himawari-8 AHI" by Aoqi Zhang et al., Earth Syst. Sci. Data Discuss., https://doi.org/10.5194/essd-2021-298-RC1, 2021

General Comments:

To date, no single dataset can provide both the three-dimensional structure of precipitation and the relevant life cycle properties. This manuscript by Zhang et al. reported on one resilient such dataset, by combining rainfall cluster and GPM 2ADPR orbital data and MCS from Himawari-8 AHI. A series of resilient reprocessing steps have proposed, including filtration, segmentation, and consolidation on the initial RCs, which makes sure the robustness and high accuracy. This method is scientifically sound, and results are pretty reasonable from the case studies shown in the present study. The dataset will definitely help facilitate three- dimensional studies of the life cycle evolution of precipitation. Therefore, I strongly recommend its acceptance after addressing the following several minor comments:

**Response**: Thanks a lot for your encouragement and helpful suggestions! The following is our replies point-by-point to your issues (presented in blue color in the **"track change file"**).

**Minor comments:**

L44-45: I noticed the references cited here are not involved in investigation of FY-4. The authors can refer to Wang et al. 2019 (doi: 10.1109/TGRS.2019.2923247) and the references therein.

**Response**: Thanks a lot for your kind reminder. We have added relevant references in the manuscript **[Line 47-48]**.

L105-106: It is not clear to me what does it mean by "the central area". Also, "in the region of subpolar westerlies with strong high-level westerly winds" is suggested to be further justified.

**Response**: Sorry for our unclear words. 1) We have changed "the central area is …" to "the RCs are …" **[line 110]**. 2) We have provided the seasonal wind field at 500 hPa **[Fig. 1]**.

Figure 6 caption: The authors can clarify what do the colored profiles in panel c-d represent.

**Response**: Thanks. We have clarified it in the figure caption **[Fig. 6]**.

Figure 7a and Figure 7b: The y-axis can be revised to the same value range, which will help the readers better understand the difference between zonal and meridian speed. Again, my guess is that the peak speed observed within 20 – 30 N could be due to the subtropical jet stream. Can the authors show the seasonal wind field at 500 – 200 hPa?

Response: Thanks a lot for your constructive suggestions. 1) we have revised the y-axis **[Fig. 7a & 7b]**.

2) We agree with you that the peak speed observed within 20–30 N could be due to the subtropical jet stream **[Line 200]**. We have provided the seasonal wind field at 500 hPa **[Fig. 1]**. The 200 hPa wind field is quite similar with 500 hPa wind field **[Fig. S1]**.

[Figure]

**Figure S1**: The same as Figure 1, but for 200 hPa wind field.

L244: it is needed to clarify what profiles are compared when the authors state "relatively consistent DSD profiles".

**Response**: Sorry for our unclear words. We have reorganized this sentence **[line 259-261]**.

Technique correction:

L237: Is "as" missing between "different" and "a result of"?

**Response**: Thanks **[Line 254]**.

L242: "corresponding" -> "correspond"

**Response**: Thanks **[Line 258]**.

The captions of Figures 3 and 5: "parts (c) and (d)" -> "panels (c) and (d)"

**Response**: Thanks a lot for your nice correction **[Fig. 3 & 5]**.

---

## Author Comment (AC2)

**Comment on essd-2021-298**

**Anonymous Referee #2**

Referee comment on "Resilient dataset of rain clusters with life cycle evolution based on observations from the GPM DPR and Himawari-8 AHI" by Aoqi Zhang et al., Earth Syst. Sci. Data Discuss., https://doi.org/10.5194/essd-2021-298-RC2, 2022

The authors present an interesting and useful data set of precipitation tracks for MCSs in Eastern Asia. The manuscript is very well written, but lacks some necessary details. I have more concerns about the data set, which is not properly described.

Response: Thanks a lot for your encouragement and helpful suggestions! The following is our replies point-by-point to your issues (presented in blue color in the **"track change file"**).

Major comments:

1. The dataset: Neither the metadata or the readme file explains what the "past", "present", and "future" suffixes actually mean. Why can, e.g., life_past be zero all the time? I would assume it would mark the hours the track has persisted. I am obviously wrong, but this is an example of where documentation is needed to make sure the data is correctly used and interpreted. Also, the time stamp is missing in all files. I encourage the authors to use the CF-convention format of the time vector.

**Response**: Thanks a lot for your constructive suggestions! **We have carefully revised the readme file and dataset. The latest version of the dataset is available at https://doi.org/10.5281/zenodo.6198716 (Zhang et al., 2022).**

1) Sorry for our carelessness. We have explained the "past", "present", and "future" suffixes in the Readme file and metadata. The life-cycle information of MCS was stored in three portions with different suffixes including "past", "present" and "future". We tracked both backward and forward the MCS with a time interval of 1 hour. The "present" suffix indicates the unique time of MCS within $\pm30$ min of the corresponding RC. The "past" suffix indicates backward track times before the "present" time, and the "future" suffix indicates forward track times after the "present" time.

2) "life_past" indicates backward life time of the corresponding MCS track, and

"life_future" indicates forward life time of the corresponding MCS track. The total life time of MCS track (units: hour) equals life_past + 1 + life_future. Note that numerous MCS tracks are short-term tracks with backward (or forward) life time less than 1 hour, the "life_past" (or "life_future") of many MCS tracks are stored as 0.

3) We have provided the time stamp for each file and used the CF-convention format of the time vector in the latest dataset.

2. The manuscript provides the spatial and temporal limitations of the data set at first on L103. This should be clear from the title, abstract and introduction. Space: eastern Asia; time: April to June 2016-2020.

**Response**: Thanks for your nice suggestion. We have clarified the spatial and temporal limitations of the data set in the title, abstract and introduction **[Line 1-2, 21-23, and 80-81]**.

3. Some details of the processing are not explain in sufficient detail. See comments below.

**Response**: Thanks for your helpful suggestions! We have added these details in the manuscript.

Detailed comments:

L17-20: Please rewrite this sentence. It is too long, not clear about how the different data sets are employed, and has some grammar issues ("tracking algorithm" should be "tracks").

**Response**: Sorry for our unclear descriptions. We have reorganized these words **[Line 18-21]**.

L103: The data set ends in 2020, which is hardly present day. Please explain why the geographical domain is not the complete disk of the geostationary statellite. Would it be technically possible to do the full domain (not considering the work of course).

**Response**: Thanks. We have corrected the unsuitable words **[Line 108]**. In the present

version, the derived dataset only covers the eastern Asia. This is because that the researches are familiar with the precipitating systems in eastern Asia. In our future work, the spatiotemporal coverage of the dataset will be further expanded to the full disk of Himawari-8.

L117: Was the contours made again on the new grid? Or were the contours remapped? Figure 4b indicates that the MCSs were remapped, rather than the DPR. Please clarify.

**Response**: Sorry for our unclear descriptions. We have corrected them in the manuscript.

1) We compared the remapped contours of the initial RCs and the contours of MCSs to determine the mapping relationships between them. Specifically, we remapped the DPR pixels of initial RCs to (0.05° × 0.05°) grids (consistent with the MCSs) and determined the overlapping grids between the initial RCs and the MCSs at the nearest time (±30 min) **[Line 123-125]**.

2) Figure 4b indicates the pixels of RCs after the two steps of RC segmentation algorithms **[Fig. 4]**.

L120: Please specify from which point you calculate the 100 km. From the nearest pixel of the contour, or the centroid of the contour? Please also explain how the centroid of the contour is defined (here and in the data set). E.g., does the centroid need to lie within the contour (comparing to a banana shape area where it might end up outside the banana).

**Response**: Thanks. 1) The distance was calculated from the nearest pixel of the contour **[Line 127]**.

2) The dataset provides the central latitude and longitude of RC and MCS. The central latitude and longitude are defined as the average of all RC pixels (or MCS pixels), so the centroid may lie outside the contour **[Line 240-241 & readme file]**.

L138: Please explain what is meant by "restore the overlapping grids".

**Response**: Sorry for our unclear words. We have corrected them **[Line 144-146]**. In the

previous algorithms, we gridded the DPR pixels to (0.05° × 0.05°) grids to obtain the overlapping grids between the initial RCs and the MCSs. The first step was to remapped the overlapping grids to the DPR pixels using just the reverse method.

L142: What is meant with "gain weight", and "round by round"? Do you mean "we iteratively increase the RC cores by adding pixels around the area until..."

**Response**: Yes, thanks a lot for your nice suggestion **[Line 150-151]**!

L144: How is the "smallest rain rate" defined? From which relative point in space? Is the gradient determined using two points, or more?

**Response**: Sorry for our unclear words. We have reorganized these sentences **[Line 151-155]**.

If, at a certain round of the collision process, one certain pixel was allocated to multiple RC cores, its nearest 8 DPR pixels, including non-precipitation pixels, must contain allocated precipitation pixels from different RC cores. We would then calculate the rain rate gradients between the certain pixel and allocated precipitation pixels using rain rate difference divided by distance, and the certain pixel would be reallocated the RC core with minimum rain rate gradient.

L225: Please add some information about the final data set. How many tracks, which information is in the files etc.

**Response**: Thanks very much for your nice suggestion. We have added some information of the final dataset in the draft **[Line 236-240]**.